# Effect of graphene addition on tensile, flexural, and hardness behavior of GFRP composites

Siraganahalli N. Nagesh[1*], Chamarajanagar G. Ramachandra[2],
Pallagatte C. Aruna Kumara[1], Praveen Kumar Kanti[3], Chander Prakash[4],
Sandeep Kumar[5], Gabr Goshu Syum[6, 7*], Archana Bhat[8*]

1 Faculty, Department of Mechanical Engineering, Ramaiah Institute of Technology, Bengaluru, India,
2 Faculty, Department of Mechanical Engineering, Presidency University, Bengaluru, India, 3 Centre
for Research Impact & Outcome, Chikara University Institule of Engineering and Technology, Chitkara
University, Rajpura, Punjab, India, 4 Research and Innovation Cell, Rayat Bahra University, Mohali,
Punjab, India, 5 Research and Innovation Cell, Bahra University, Waknaghat, India, 6 Centre of Studies
in Oil and Gas Engineering and Technology(CS-OGET), Eduardo Mondlane University, Bairro Luís Cabral,
Maputo, Mozambique, 7 Faculty of Mechanical and Industrial Engineering, EiT-M, Mekelle University,
Mekelle, Tigray, Ethiopia, 8 Nitte (Deemed to be University), NSAM First Grade College (NSAMFGC),
Department of Computer Science, Udupi, India

* gabr.goshu@mu.edu.et (GGS); snnagesh80@gmail.com (NSN); archanabhat5@nitte.edu.in (AB)

Cheng University, Taiwan & Australian Center
for Sustainable Development Research and
Innovation (ACSDRI), AUSTRALIA

**Peer Review History:** PLOS recognizes the
benefits of transparency in the peer review
process; therefore, we enable the publication
of all of the content of peer review and
author responses alongside final, published
articles. The editorial history of this article is
available here: https://doi.org/10.1371/journal.
pone.0340894

## Abstract

Polymer Matrix Composites represent a new generation of engineering materials in
which matrix materials are altered by reinforcing filler materials to enhance strength
and other properties. In the present work, the Glass fiber reinforced polymer (GFRP)
composites were fabricated by adding different compositions of Graphene as a Nano
filler material using the Hand layup technique, followed by the Vacuum bagging pro-
cess. The Graphene filler material was varied from 0 wt. %, 2 wt. %, and 4 wt. %. The
prepared composites were characterized by subjecting them to various mechanical
tests such as tensile test, flexural test, and micro hardness test. The experimental
results revealed an increase in tensile strength by 5.48%, flexural strength by 27.74%,
and a reduction in microhardness by 20% for 4 wt. % filler added GFRP compared to
neat GFRP, indicating reduced brittleness and enhanced ductility. Furthermore, addition
of graphene resulted in a 9% increase in elongation and improved interfacial bonding
between glass fiber and epoxy matrix was observed. Scanning electron microscopy
(SEM) analysis confirmed uniform graphene dispersion, reduced voids, and improved
fiber–matrix adhesion. The results demonstrate that the inclusion of 4 wt. % graphene
filler achieves an optimal balance between strength, flexibility, and interfacial bonding,
making these composites suitable for structural and lightweight applications.

## 1. Introduction

Materials are possibly more deeply ingrained in our everyday lives than we realize.
Housing, transportation, communication, food production, and defense, all aspects

**Data availability statement:** All data underlying the findings of this study are fully available within the manuscript and its Supporting Information files. Due to institutional restrictions and confidentiality agreements, additional raw data cannot be publicly deposited. Data requests may be submitted to the Research Ethics office and Data Access Center at the Centre of Studies in Oil and Gas Engineering and Technology (CS-OGET), Eduardo Mondlane University, Maputo, Mozambique (email: csoget.geral@uem.mz), subject to ethical review and approval.

**Funding:** The author(s) received no specific funding for this work.

**Competing interests:** The authors declare no competing financial interests or personal relationships that could influence this work.

of our daily existence, are influenced, in one way or another, by materials [1,2]. Historically, the development of societies has reflected their ability to produce or manipulate materials to meet their needs and the level of their material advancement. Furthermore, it has been discovered that the properties of materials can be altered by adding or combining them with other materials or by heat treatment to produce superior attributes in the basic material. The demand for materials is increasing across industries to establish the sectors with enhanced properties [3], reduced costs [4], and improved sustainability [5]. Composites are created by utilizing constituent materials, namely the matrix and reinforcement [6]. The function of the matrix is to envelop and support the reinforcement materials in their relative positions, imparting specific mechanical and physical properties. Composites are gaining importance in various fields such as automotive, aerospace, defense, biomedical, electronic equipment, and sports.

In the current scenario, polymers play a major role in many applications due to their unique properties, such as lightweight, chemical stability [7]. It can be processed easily and obtained at a lower cost, gaining its importance in many aspects of human life. The properties of the polymers can be altered suitably using fibers, organic/ inorganic particles, and nanofillers. These materials have lower density and differ from ceramics and metals in terms of strength and stiffness. Polymers can be molded into complex shapes as they exhibit elastic and ductile behavior [8]. Polymer matrix composites mainly comprise epoxy resins as the matrix materials, gaining importance for structural engineers due to their well-balanced balance of mechanical and chemical properties [9–12]. The material also offers wide flexibility and processing versatility. It exhibits superior adhesion to various reinforcement materials, lower shrinkage, and high strength, making it suitable for various applications [9,13,14]. The research on the development high performance resins with various inclusions of reinforcements such as carbon fiber [15,16,17], glass fiber, nanofiber [18], Nano particles etc. leads to the significant progress in the advancement of polymer matrix composites. Glass fibers are used as reinforcement materials in structural applications due to its specific strength and recent developments in the fiber reinforced plastic materials [12,19–23]. Fiber reinforced polymer composites show high stiffness and strength, making them suitable in various fields like aerospace industries, concrete structures, automotive industries, and wind turbines, owing to their mechanical, thermal, and chemical properties [24–27,28]. The interface between the glass fiber and polymer matrix has some control over certain mechanical properties [29,30]. The strength of the interface achievement is the key issue, and the adhesion at this junction becomes weak due to poor adsorption and wettability [31]. The major drawback associated with the usage of FRP in weak parts of the component leads to failure and damage as it is subjected to different loading conditions. The properties of FRPs can be improved significantly by the addition of nanoparticles, which improves the stability and durability of the components [32]. Graphene is one of the two-dimensional layered effective carbon nanoparticles used as a filler material in polymer composites, exhibiting excellent mechanical, thermal, and electrical properties along with a high surface area [32–36]. Addition of graphene to the polymer matrices yields a modest volume fraction,

enhancing certain polymer properties. Graphene added nanocomposites gain importance in the development of novel materials in the domain of alternative energy sources. It exhibits superior performance in the development of lithium–ion batteries, electrodes, and solar cells [37]. Many researchers reported that the addition of nanoparticles below 1 wt. % as reinforcement leads to no significant change in the results while, higher content of nanoparticles above 5 wt. % identified two major causes: one problem with the uniform dispersion of nano filler in the matrix. Another problem with interfacial adhesion of the filler in the matrix [13,21,38].

Recent investigations on hybrid glass fiber composites reinforced with particulate fillers have demonstrated noticeable improvements in mechanical and tribological behavior. For instance, Mechanical and tribological characteristics of glass fiber and rice stubble-filled epoxy–LD sludge hybrid composites reported that hybrid fillers enhanced wear resistance and load-bearing capacity through improved interfacial bonding [39]. Similarly, Mechanical and tribo-performance analysis of Linz Donawitz sludge-filled glass–epoxy composites using Taguchi experimental design highlighted the influence of optimized filler ratios on flexural strength and wear performance [40]. However, despite these promising developments, most studies have focused on micro-scale or waste-derived fillers, and limited attention has been given to the effect of graphene nanoparticle reinforcement on the mechanical and interfacial characteristics of GFRP composites. Hence in this study, an attempt will be made to study the effect of graphene powder as filler material in larger variations and glass fiber as base material. In this study, we are focusing on mechanical properties and characterization of glass fiber reinforced epoxy polymer composites with variation in filler material as graphene powder in larger constituents. The prepared specimens will be subjected to different tests to investigate their mechanical properties, and the dispersion quality will be examined through a scanning electron microscope.

The novelty of the present research lies in selection of correct combination of graphene filler concentration to achieve a balance between improved tensile and flexural properties. Some of the previous studies that focused solely on mechanical enhancement, this work emphasizes the interfacial microstructural improvements achieved through a controlled dispersion process involving thermal-assisted sonication and vacuum compaction. The systematic comparison of 0 wt. %, 2 wt. %, and 4 wt. % graphene filler content provides new insights into the synergistic effects of nanoparticle reinforcement on load transfer efficiency, ductility, and void minimization. The resulting composite formulation offers a cost-effective and scalable approach for developing next-generation lightweight structural materials.

## 2. Materials and methods

### 2.1. Experimental detail work

The approach for manufacturing the composite materials in this study was carried out using a hand lay-up process, followed by a vacuum bagging process to remove the air trapped between the layers of glass fiber and epoxy matrix [13,41]. The reinforcement material, specifically glass fiber, was kept constant at 60 wt. %, while the filler material, specifically Graphene powder, varied from 2 wt. % to 4 wt. % by reducing the epoxy matrix by the same amount. A plywood board measuring 400 x 400 mm was used, and plastic tape was placed on the upper surface to ensure it acted as a releasing agent. In this work, an open mold made from plywood was considered a substitute for a metal mold or any other type of mold, as it is inexpensive, easy to handle, and lightweight. The required quantities of epoxy, hardener, glass fiber, and graphene powder were weighed using an electronic scale and placed in separate beakers. The mixture of epoxy and hardener was stirred properly employing a mechanical stirrer for a duration of 5 minutes. For the graphene added composites, a combination of mechanical stirring, thermal pre-treatment, and ultrasonic sonication was employed to ensure the uniform dispersion of graphene powder within the epoxy matrix. Initially, the graphene powder was added to the pre-heated epoxy resin at 70 °C to facilitate wetting and breakdown of agglomerates. The mixture was allowed to cool at room temperature at a slower rate by partially heating it. Sonication was performed for about 20 minutes at 150 V; then, the hardener was added, and stirring continued with the mechanical stirrer for another 5minutes. This dual approach of sonication and thermal mixing promoted homogeneous distribution of graphene nanosheets throughout the

epoxy matrix, minimizing particle agglomeration and enhancing filler–matrix interaction. A portion of the prepared matrix material was poured into the mold, followed by a layer of glass fiber mat, with more matrix material poured on top. This process was repeated until the desired thickness of 2 mm was achieved. Once the required thickness of the composite material was reached, the upper layer was covered with plastic tape to serve as a releasing agent, and excess trapped air was removed using a roller. The vacuum bagging process was then conducted. Specimens were cut according to ASTM standards using the water jet cutting technique. A tensile test was performed according to ASTM D3039 standard [42] with a Mecmesin multi-tester machine in Fig 1 using a dog bone-shaped specimen. The test was conducted at a constant loading speed of 2 mm/min and a strain rate of 0.1 mm/min with a span length of 110 mm. The dimensions of the specimen were prepared as per ASTM D3039 standards, as shown in Fig 2. Tensile specimens of various combinations are displayed in Fig 3 before and after testing.

Flexural test was carried out according to ASTM D790 standard [43] with a Mecmesin multi tester machine using a rectangular-shaped specimen. The test was performed at a constant loading speed of 2 mm/min with a span length of 110 mm. Dimensions of the specimen were prepared according to ASTM D3039 standard [44], shown in Fig 4 below.

Micro Hardness test was carried out according to ASTM E384 standard with Matsuzawa micro-hardness tester (Fig 5) Model MMT-X7A using a square-shaped specimen shown in Fig 6.

The morphology of carboxyl graphene/epoxy composites was investigated to determine the dispersion of carboxyl graphene powder in the epoxy matrix using a scanning electron microscope (SEM).

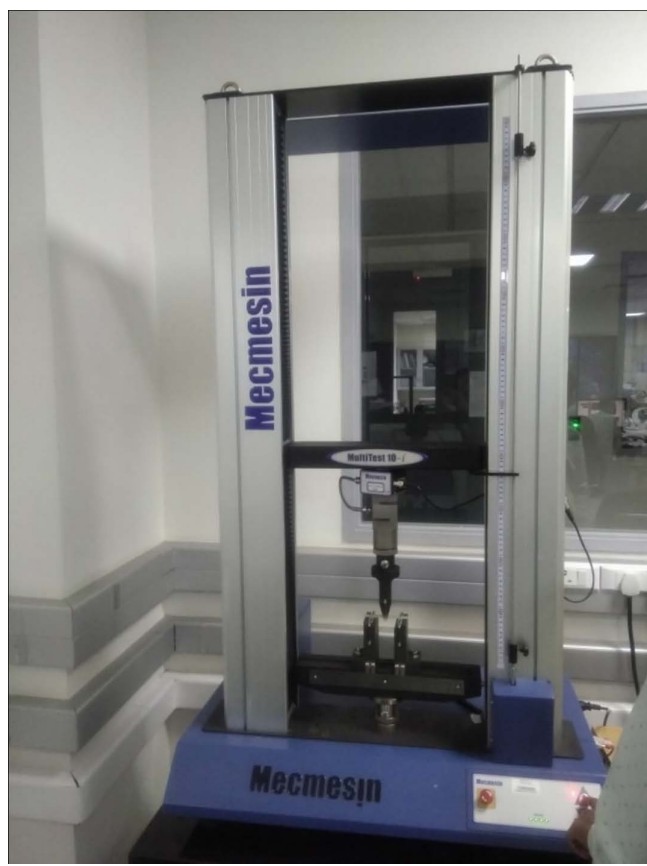

**Fig 1. Mecmesin Multi tester.**

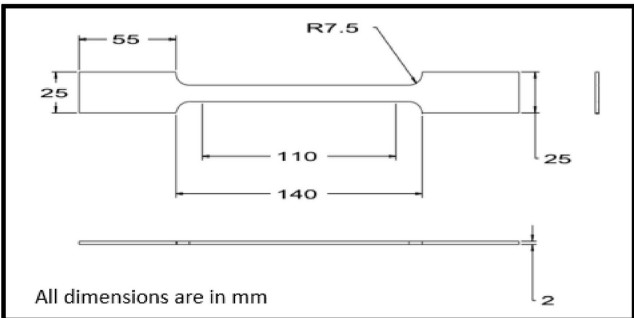

**Fig 2. Tensile test specimen dimensions.**

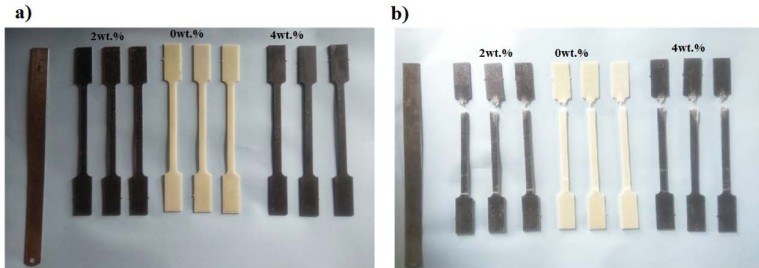

**Fig 3. Tensile test specimens with and without filler, (a) before testing, (b) after testing.**

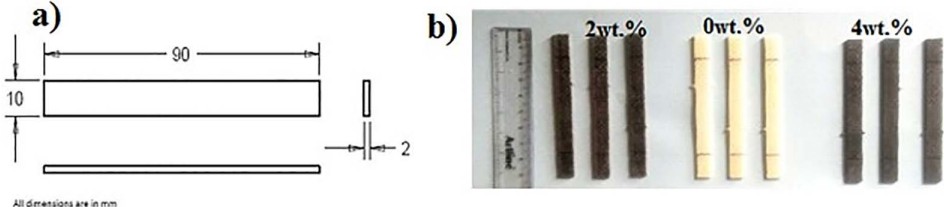

**Fig 4. Flexural test specimens.**

## 3. Results and discussion

### 3.1. Results discussion on tensile testing

The tensile stress–strain diagram of the GFRP composites with varying graphene content (0 wt.%, 2 wt.%, and 4 wt.%) are shown in Fig 7. GFRP specimens exhibits by an initial linear-elastic region followed by a gradual deviation from linearity prior to failure. The plain composite (0 wt. %) shows a predominantly linear response up to the ultimate stress, indicating a brittle failure mode governed by fibre fracture. In contrast, the 2 wt. % and 4 wt. % graphene-filled composites show a noticeable nonlinear region before the ultimate tensile strength (UTS), indicating progressive matrix yielding, fibre–matrix debonding, prior to rupture. Although a distinct yield point, as observed in metallic materials, is not present in fibre-reinforced polymers, the onset of nonlinearity in the 2 wt. % and 4 wt. % composites signifies a higher strain tolerance and delayed catastrophic failure. This behavior leads to a slight reduction in hardness corresponds to improved deformability of the matrix and enhanced ductility of the composite system.

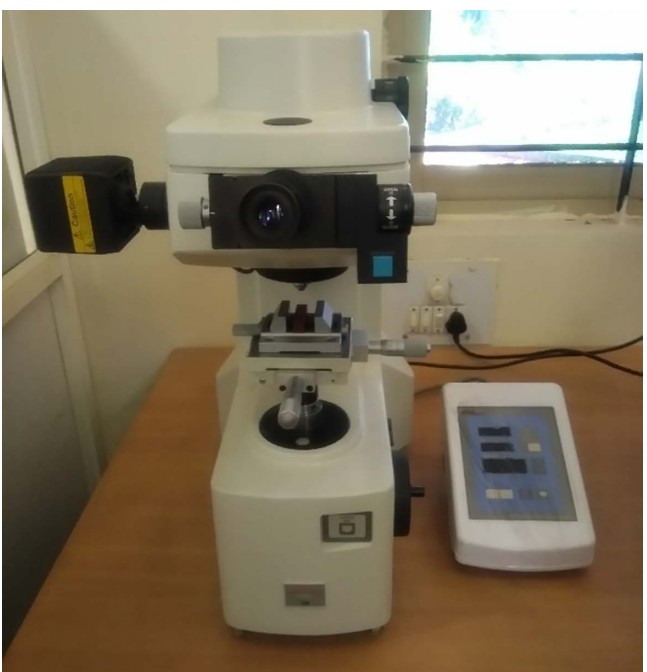

**Fig 5. Matsuzawa micro-hardness tester.**

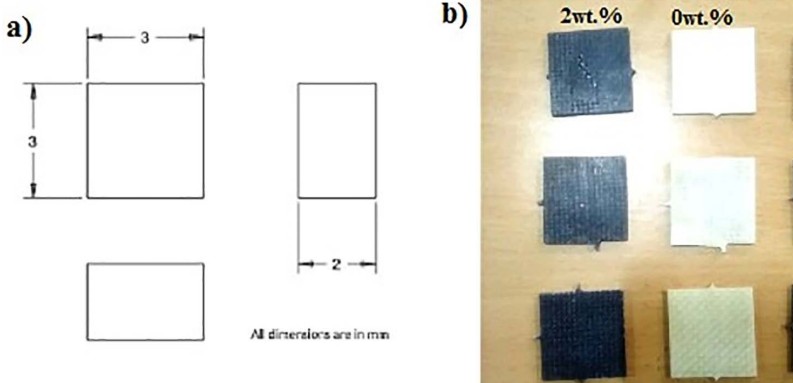

**Fig 6. Micro Hardness test specimens.**

The inclusion of graphene improves interfacial adhesion and energy dissipation capability, allowing limited plastic deformation before complete failure, consistent with reported matrix work hardening behavior in other alloy-based systems [45]. The stress–strain curves thus confirm the ductility enhancement associated with graphene reinforcement while maintaining the strength characteristics of the GFRP composites.

Fig 8 depicts the maximum tensile load v/s maximum displacement for all the combinations of composites.

- It was observed that the load-carrying capacity was increased with an increase in Graphene content in the composites. Initially, the tensile load obtained for the plain composite without graphene powder was 7587.66 N, and it increased to 8004 N for the composite with 4 wt. % of Graphene content.

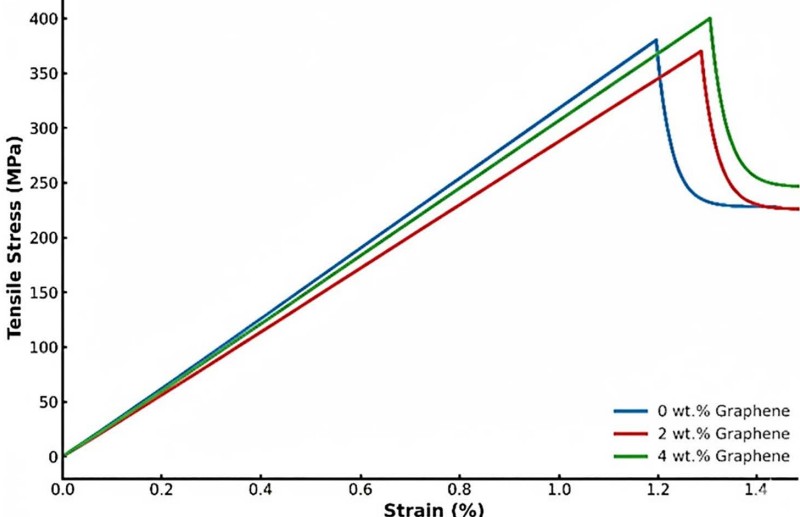

**Fig 7. Stress strain curve of the different filler added Composites.**

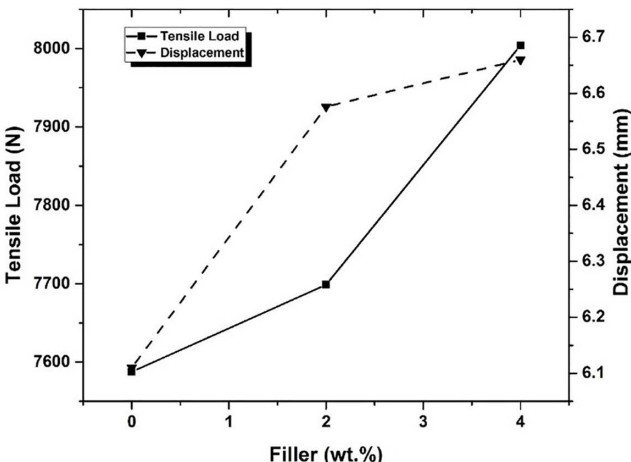

**Fig 8. Tensile load vs. Displacement for different compositions of filler.**

- The tensile strength of composites depends on interfacial bonding strength between matrix and reinforcement to a larger extent, and also on the inherent properties of composite ingredients [46]. It was observed that there is a 1.46% tensile load and 7.63% of elongation improvement for composites with 2 wt. % filler material, while the composite with 4 wt. % filler material showed 5.50% tensile load and 9% elongation improvement.

- The role of glass fibers in the composite limits the failure [47] and the increase in the filler material content exhibits the upward trend in tensile properties [34]. Fig 9 shows the marginal increase in the ultimate tensile strength as a result of increased interfacial bonding between the glass fiber and epoxy matrix due to the addition of Graphene as filler material.

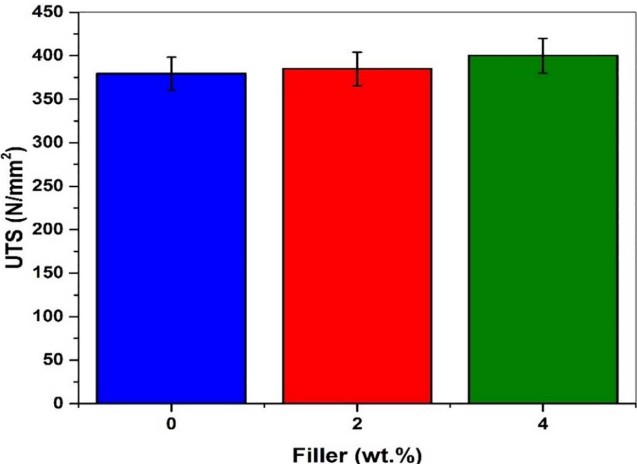

**Fig 9. Ultimate tensile strength of filler-added composites.**

### 3.2. Results discussion on flexural test

- The comparative plot of the flexural load versus displacement for each of the combinations of glass fiber reinforced composites is shown in Fig 10. It is observed from the graph that the flexural load-carrying capacity was increased with an increase in Graphene content in the composites.

- The improvement of flexural load from 149 N for plain GFRP to 190.33 N for GFRP with 4 wt. % filler material is a result of good adhesion with the matrix formed with the addition of Graphene powder in the material [13]. It is seen that there is 17.68% of flexural load and 5.38% of elongation improvement for composites with 2 wt. % filler material, and the composite with 4 wt. % filler material showed 27.74% flexural load and 10.13% of elongation improvements obtained from the flexural three-point bending test.

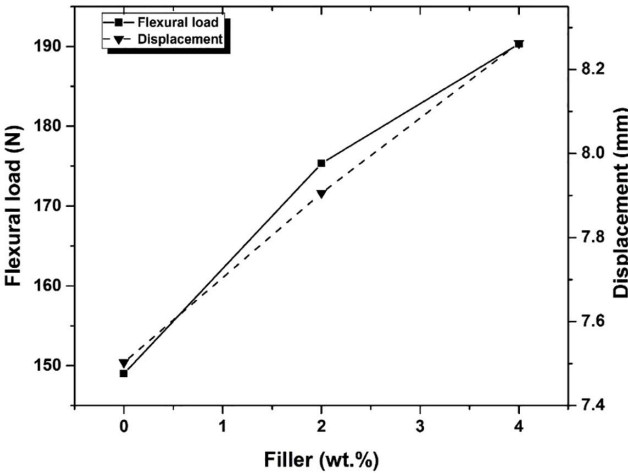

**Fig 10. Flexural load vs. Displacement for different compositions of filler.**

- Fig 11 indicates that the flexural strength of filler-added GFRP was more compared with the Normal GFRP. This is due to the uniform dispersion of filler material in the matrix enhances the flexural properties of the materials by increasing interfacial bonding strength, ensuring higher reliability and flexural strength in hybrid composite system [15,48,49].

The mechanical properties of the investigated samples are summarized in Table 1, which presents the results of the tensile tests, including ultimate tensile strength (UTS), yield strength (YS), and percentage elongation. As an increase in tensile strength and elongation was seen because of the addition of graphene filler, with the highest improvement at 4 wt. % filler content (5.5% improvement in tensile strength and 9% in elongation). This may be due to the interfacial bonding between the glass fiber and epoxy matrix existing strongly together with a suitable dispersion of graphene.

$^2$ show a more pronounced increase with graphene addition at 4 wt. % filler loading (Load increase of 27.74%, elongation increase of 10.13%). This is due to the uniform distribution of the filler that will provide good matrix adhesion and suitable stress transfer. Finally, the graphene filler enhances the mechanical properties of GFRP composites, with maximum property enhancement shown at 4 wt. % filler loading.

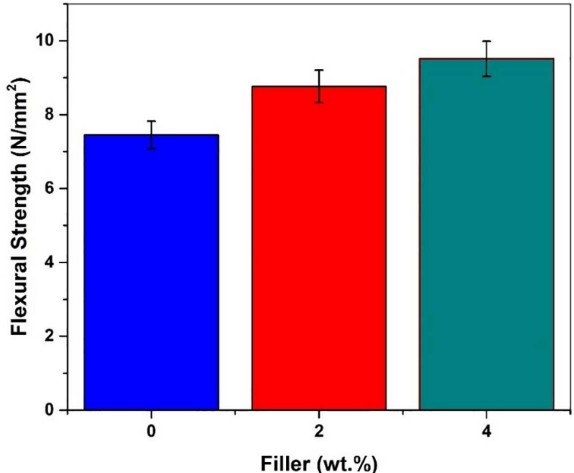

**Fig 11. Flexural strength of filler-added composites.**

**Table 1. Summary of Mechanical Test Results.**

| (Observation) | Filler Content (wt. % Graphene) | Plain Composite (0 wt. %) | 2 wt. % Filler | 4 wt. % Filler | Improvement (2 wt. % vs. 0 wt. %) | Improvement (4 wt. % vs. 0 wt. %) |
|---|---|---|---|---|---|---|
| Tensile Load (N) | – | 7587.66 N | – | 8004 N | +1.46% | +5.50% |
| Tensile Elongation (%) | – | (0 wt. % graphene filler). | +7.63% | +9% | – | – |
| Ultimate Tensile Strength | – | – | Borderline increase | Borderline increase | Improved interfacial bonding | Improved interfacial bonding |
| Flexural Load (N) | – | 149 N | – | 190.33 N | +17.68% | +27.74% |
| Flexural Elongation (%) | – | Baseline | +5.38% | +10.13% | – | – |
| Flexural Strength | – | Lower | Higher | Highest | Enhanced dispersion | Enhanced dispersion |

## 3.3  Results for micro hardness test

Fig 12 shows the average hardness values of different materials with and without filler added. The graph revealed that the hardness values of the GFRP composites gradually reduce with an increase in Graphene content.

**a)** GFRP composites with 2 wt. % filler material exhibit an 8.99% decrease in hardness values, whereas GFRP composites with 4 wt. % filler material show a 20% decrease in the values of hardness. It is revealed from the experimental results that the addition of Graphene powder resulted in a decrease in the brittleness of the composites.

## 3.4  SEM characterization

SEM analysis was carried out on GFRP composites and filler-added GFRP composites at 1000X and 5000X magnification shown in Fig 13 below. The study reveals the internal structure of the composites, specifically the arrangement of fibers in the matrix, fiber pullout, and voids in the structure.

1. For the 0 wt. % composite. In contrast, the 2 wt. % and 4 wt. % graphene-filled composites exhibit uniform filler dispersion within the epoxy matrix and improved fiber–matrix adhesion. The graphene nanosheets act as micro-bridges at the interface, improving load transfer efficiency and delaying crack initiation.

2. GFRP composites show the fractured glass fibers on the smooth fractured surface show interfacial gaps, fiber pullouts, and matrix cracking, indicating poor stress transfer and premature interfacial failure, implying that the adhesion between the glass fiber layers and the resin is weak. The adhesion between the glass fiber and the matrix was observed to be stronger with the addition of Graphene powder, as the fibers are bonded together more firmly. The composite with 4 wt. % Graphene powder exhibits fiber pullouts in a single plane as a result of uniform pullout and more intact fiber adhesion.

3. The filler material 2 wt. % and 4 wt. % graphene-filled composites exhibit uniform filler dispersion within the epoxy matrix and improved fiber–matrix adhesion. The graphene nanosheets act as micro-bridges at the interface, improving load transfer efficiency and delaying crack initiation. Graphene, dispersed uniformly in the matrix, revealed at higher magnification, 5000X. This gives hints of minimum cluster and agglomeration of Graphene powder in the matrix, resulting in strong interfacial bonding between the matrix and reinforcement. GFRP with 4 wt. % filler material revealed minimum void formation in the matrix. The fractured surface of the tensile specimens of the composites without the

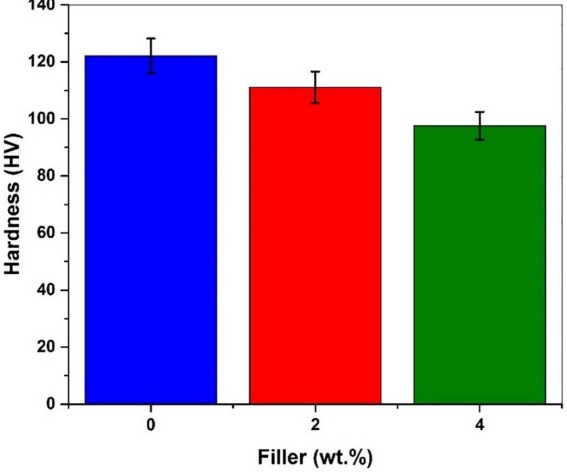

**Fig 12.  Hardness values of filler-added composites.**

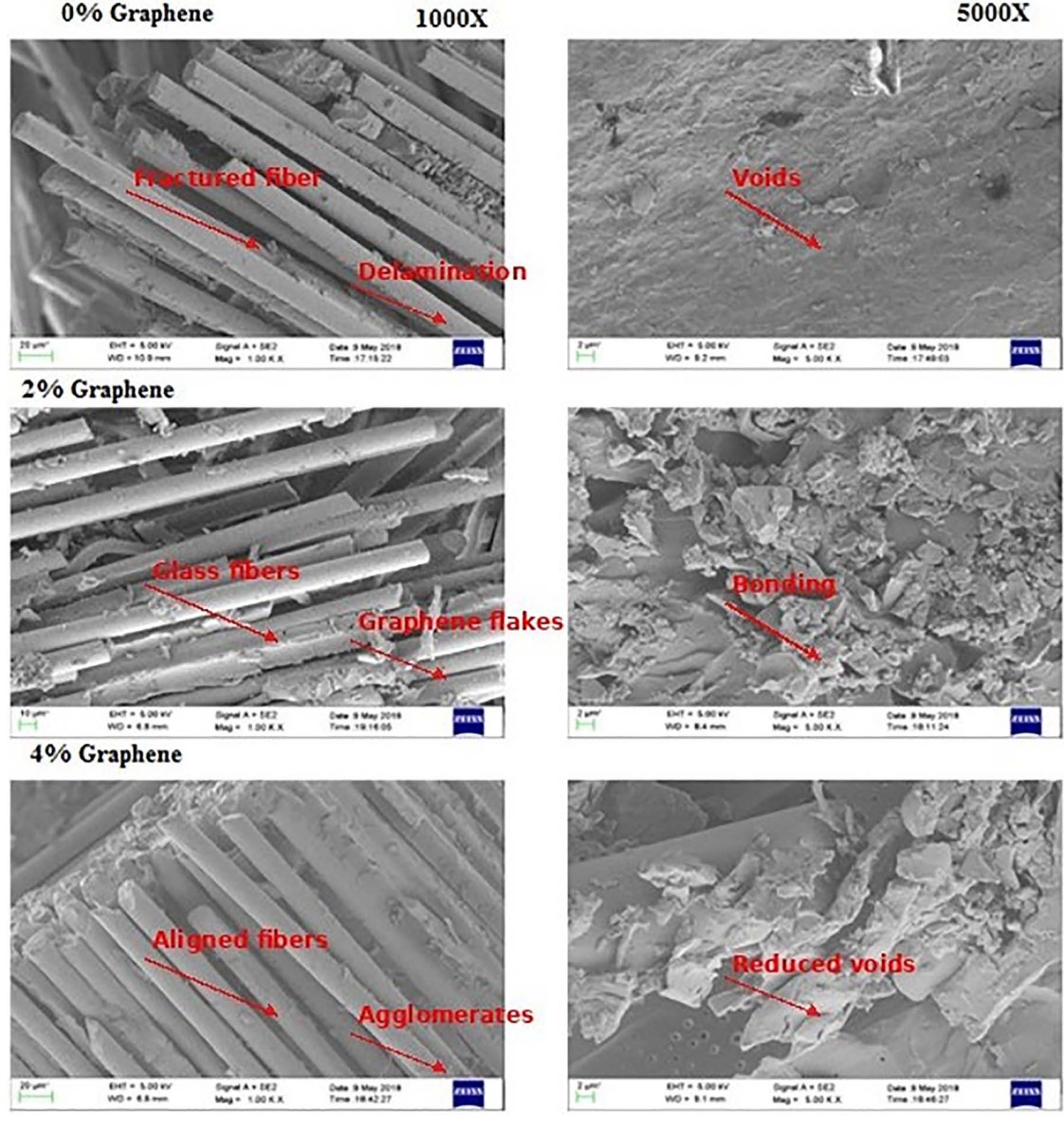

**Fig 13. SEM images of GFRP and filler added to GFRP composites.**

filler materials revealed predominant delamination caused due to the interaction between the glass fiber and the matrix material.

4. The delamination stress at the fractured surface accelerates fracture at the matrix and reinforcement interface, resulting in complete fracture at the surface. From the SEM micrograph, fiber pulls out were predominant in the composite without the filler material as a result of higher displacement, whereas the filler added composites show some hindrance to fiber pullouts, resulting in higher load bearing capacity of composites. Hence, the filler material reduces the interfacial interaction between the matrix and reinforcement. At higher graphene reinforced composites (4 wt. %), the network of graphene platelets reinforces the matrix by restricting polymer chain mobility and hindering crack propagation through deflection

and branching. These microstructural mechanisms explain the increase in tensile and flexural strength and the slight decrease in hardness, which collectively indicate a transition toward a more ductile failure mode. The uniform Graphene distribution also facilitates energy dissipation during deformation, contributing to enhanced toughness and strain accommodation consistent with similar findings on Graphene nanoplatelet dispersion behavior in metallic matrices [50].

The results of microhardness measurements and SEM characterization are presented in Table 2. According to results, the hardness of GFRP composites decreased on raising the increase in graphene content in the composites (8.99% at 2 wt. %, 20% at 4 wt. %), thus implying reduced brittleness. This means that graphene increases the ductility of the composites but decreases the surface hardness. Simultaneously, Findings were observed in interfacial fiber-matrix bonding in the composites and found that Plain GFRP shows weak interfacial fiber-matrix bonding that favors fiber pullouts and delamination voids, leading to premature failure; meanwhile, the filler-added composites (2 wt. % & 4 wt. % graphene) yielded better interfacial adhesion with well-dispersed graphene and fewer voids. The 4 wt. % filler exhibited fiber pull-outs in one plane, which confirms enhanced load transfer and matrix-reinforcement bonding.

## 4. Conclusions

The present study investigated the influence of graphene powder addition on the mechanical and microstructural behavior of Glass Fiber Reinforced Polymer (GFRP) composites fabricated using the hand lay-up method followed by vacuum bagging. The experimental results demonstrated that the inclusion of graphene significantly enhanced the composite performance depending on the filler content. The 4 wt. % graphene-added composite showed the maximum improvement in tensile and flexural behavior, with an increase of 5.48% in tensile strength and 27.74% in flexural strength compared to the plain GFRP composite. These improvements are primarily attributed to enhanced interfacial adhesion between the glass fibers and epoxy matrix, as confirmed by SEM analysis, which revealed uniform graphene dispersion, fewer voids, and restricted fiber pullout. Furthermore, the microhardness decreased by 20% with the addition of 4 wt. % graphene, indicating reduced brittleness and improved ductility of the composite. This reduction in surface hardness suggests that graphene facilitates better energy absorption under deformation, which is beneficial for applications requiring impact resistance.

The enhanced graphene-modified GFRP composites are well-suited for aerospace structures, automotive body panels, wind turbine blades, marine applications, and sports equipment, where a high strength-to-weight ratio, durability, and improved fatigue resistance are essential. The study also provides a foundation for future research on optimizing Nano filler dispersion techniques, scaling the fabrication process for industrial applications, and exploring hybrid filler systems to further tailor the multifunctional properties of advanced composite materials.

## Abbreviations

For ease of understanding and consistency, Table 3 lists the common abbreviations used throughout this manuscript. These abbreviations represent frequently used terms related to materials, testing methods, and coating processes, ensuring clarity and reducing repetition within the text.

**Table 2. Result Summary of Microhardness and (SEM Characterization).**

| (Observation) | (0 wt.% Filler – Plain GFRP) | (With Filler – 2 wt. % & 4 wt. % Graphene) |
|---|---|---|
| **Microhardness** | Higher (Baseline) | 2 wt. % filler: 8.99% decrease<br>4 wt. % filler: 20% decrease |
| **Brittleness** | Harder | Reduced brittleness with filler addition |
| **Fiber-Matrix Adhesion (SEM)** | Weak adhesion, fiber pullouts, delamination | Stronger adhesion, uniform fiber bonding, fewer voids |
| **Fiber Pullout Behavior** | Main pullouts, smooth fracture surface | Hindered pullouts, intact fibers (4 wt. % filler) |
| **Filler Dispersion (SEM at 5000X)** | N/A (No filler) | Uniform dispersion, minimal agglomeration |
| **Void Development** | Noticeable voids & delamination | Summary voids (especially at 4 wt. % filler) |

**Table 3. Contains some common abbreviations used throughout this manuscript.**

| Abbreviations | Meaning |
| --- | --- |
| SEM | Scanning Electron Microscope |
| ASTM | American Society of Testing and Materials |
| ASTM | American Society of Testing and Materials |
| GERP | Glass Fiber-Reinforced Polymer or Plastic |
| FRP | Fiber Reinforced Polymer |

## Supporting information

**S1 File. Experimental data supporting the results of the present study.**
(XLSX)

## Author contributions

**Conceptualization:** Siraganahalli N. Nagesh.

**Formal analysis:** Chander Prakash.

**Investigation:** Chamarajanagar G. Ramachandra, Praveen Kumar Kanti.

**Methodology:** Praveen Kumar Kanti, Chander Prakash.

**Resources:** Chander Prakash, Sandeep Kumar.

**Supervision:** Praveen Kumar Kanti.

**Validation:** Sandeep Kumar.

**Visualization:** Chamarajanagar G. Ramachandra, Praveen Kumar Kanti, Gabr Goshu Syum.

**Writing – original draft:** Siraganahalli N. Nagesh.

**Writing – review & editing:** Gabr Goshu Syum, Archana Bhat.

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
