## [Decision Letter · Decision Letter 0]

8 Oct 2025

Dear Dr. Syum,

We look forward to receiving your revised manuscript.

Kind regards,

Karthik Kannan, Ph. D.,

Academic Editor

PLOS ONE

Journal Requirements:

2. In the online submission form you indicate that your data is not available for proprietary reasons and have provided a contact point for accessing this data. Please note that your current contact point is a co-author on this manuscript. According to our Data Policy, the contact point must not be an author on the manuscript and must be an institutional contact, ideally not an individual. Please revise your data statement to a non-author institutional point of contact, such as a data access or ethics committee, and send this to us via return email. Please also include contact information for the third party organization, and please include the full citation of where the data can be found.

4. Please ensure that you refer to Figure 9 in your text as, if accepted, production will need this reference to link the reader to the figure.

5. We note you have included a table to which you do not refer in the text of your manuscript. Please ensure that you refer to Table 1, 2, and 3 in your text; if accepted, production will need this reference to link the reader to the Table.

Reviewer's Responses to Questions

**Comments to the Author**

1. Is the manuscript technically sound, and do the data support the conclusions?

Reviewer #1: Yes

Reviewer #2: Yes

Reviewer #3: Yes

2. Has the statistical analysis been performed appropriately and rigorously?

Reviewer #1: Yes

Reviewer #2: Yes

Reviewer #3: Yes

3. Have the authors made all data underlying the findings in their manuscript fully available?

Reviewer #1: Yes

Reviewer #2: No

Reviewer #3: Yes

4. Is the manuscript presented in an intelligible fashion and written in standard English?

Reviewer #1: Yes

Reviewer #2: Yes

Reviewer #3: Yes

Reviewer #1: It is interesting to review the manuscript entitled “Effect of Graphene Addition on Tensile, Flexural, and Hardness Behavior of GFRP Composites”. The language of the paper is good. The paper may be accepted after addressing the reviewer’s comments.

1. Authors need to add more quantitative results in the abstract section.

2. Consider explicitly stating the novelty of the current work toward the end of the Introduction.

3. wt.% can be written as wt. % (give a space)

4. How the uniform distribution of Graphene powder in matrix phase was ensured?

5. Paper requires considerable modification. The manuscript contains few typos which should be corrected. Some words are merged. Check the text for clarity, grammar and syntax throughout.

• 400x400 MM …………. MM must be written in small letter

• 70°C, 190.33N............. give a space between value and unit, and do similar corrections throughout the manuscript.

• GFRP with 4% filler material…. Mention wt. % or vol. %

6. Add more suitable references for the mentioned ASTM standards. Authors can cite following article;

ASTM D3039: Exploring the use of TiO2 filler in short Ficus Benghalensis natural fibre-based polymeric composites.

ASTM D790: Mechanical and tribo-performance analysis of LD sludge filled wood apple dust-epoxy composites using response surface methodology.

ASTM E384: TRIBOLOGICAL STUDY ON SLURRY ABRASIVE WEAR BEHAVIOR OF NONWOVEN VISCOSE FABRIC COMPOSITES WITH DOE APPROACH

7. In Fig 12, annotate SEM images to make it easier for readers to connect what you described in text with the visual. Authors are requested to add more description in Micro-structural Studies section.

8. How the content of filler selected?

9. Previous research on glass fiber and particulate filler needs to be described in detail to obtain research gaps so that the research position becomes clear and this topic is worthy of research. Authors can cite recent papers like;

Mechanical and tribological characteristics of glass fiber and rice stubble-filled epoxy-LD sludge hybrid composites.

Mechanical and Tribo-performance analysis of Linz Donawitz sludge-filled glass–epoxy composites using Taguchi experimental design.

10. Add technical facts to the conclusions to improve them.

11. In conclusion, be more specific in applications.

Reviewer #2: The manuscript is well written and the results well presented. However, by adding fillers and claiming that the hardness decreases and therefore the ductility increases, it is essential to investigate the stress-strain curves to see if a yield point can be observed in the specimen. I would, therefore, like to see these curves either in the manuscript or separately provided for me by the editor.

Reviewer #3: Dear Author,

Please address the following questions in detail. Minor revision is required.

1. How was the repeatability of each test ensured, and were standard deviations or error bars considered in the reported data?

2. What calibration procedure was followed for the instruments used to maintain measurement accuracy throughout the experiments?

3. Were environmental conditions such as temperature and humidity controlled during testing, and could they influence the observed outcomes?

4. What is the rationale behind the selected process parameters or test conditions—were they optimized or based on preliminary trials or literature data?

5. How was the sample preparation process standardized to avoid variation between different test specimens?

6. The paper reports improved results under specific conditions. What are the underlying microstructural or physicochemical mechanisms responsible for this improvement?

7. Were any anomalies or outliers observed in the dataset, and how were they treated during analysis?

8. Have the authors considered the influence of interfacial bonding or phase interaction in explaining the observed property enhancement?

9. Can the authors provide a mechanistic explanation linking the morphology or microstructure to the obtained performance outcomes?

10. How would the results change if the testing was performed under dynamic or real operating conditions instead of static conditions?

11. The following references are recommended for citation.

Soundararajan, R., Dharunprakash, E., Arjunkumaar, N. et al. Appraisal of Mechanical and Tribological Performance of Onyx and Carbon Fiber Composites Produced Through Various Layering Approaches in Continuous Fused Filament Fabricated. J. Inst. Eng. India Ser. D 106, 215–229 (2025). https://doi.org/10.1007/s40033-023-00626-z

Soundararajan R, Kaviyarasan K, Sathishkumar A, Muthiya Solomon J. Evaluating the impact of post-processing on the wear and friction properties of polyamide 6 carbon fiber composites produced by fused deposition modeling. Proceedings of the Institution of Mechanical Engineers, Part E: Journal of Process Mechanical Engineering. 2024;0(0). https://doi.org/10.1177/09544089241300001

**Do you want your identity to be public for this peer review?** For information about this choice, including consent withdrawal, please see our Privacy Policy

Reviewer #1: No

Reviewer #2: **Yes:** Professor Manouchehr Salehi

Reviewer #3: **Yes:** Dr.R.Soundararajan

---

## [Author Response · Author response to Decision Letter 1]

14 Nov 2025

Thank you very much for the Reviewers and Editors for this valuable comment. We have revised the manuscript accordingly.We belive that the manuscript fits with the standard requirements of PLOS ONE Journal.

---

## [Decision Letter · Decision Letter 1]

11 Dec 2025

Dear Dr. Syum,

Thank you for submitting your manuscript to PLOS ONE. After careful consideration, we feel that it has merit but does not fully meet PLOS ONE’s publication criteria as it currently stands. Therefore, we invite you to submit a revised version of the manuscript that addresses the points raised during the review process.

We look forward to receiving your revised manuscript.

Kind regards,

Karthik Kannan, Ph. D.,

Academic Editor

PLOS One

Journal Requirements:

Reviewers' comments:

Reviewer's Responses to Questions

**Comments to the Author**

Reviewer #1: All comments have been addressed

Reviewer #2: (No Response)

Reviewer #3: All comments have been addressed

2. Is the manuscript technically sound, and do the data support the conclusions?

Reviewer #1: Yes

Reviewer #2: No

Reviewer #3: Yes

3. Has the statistical analysis been performed appropriately and rigorously?

Reviewer #1: Yes

Reviewer #2: No

Reviewer #3: Yes

4. Have the authors made all data underlying the findings in their manuscript fully available?

Reviewer #1: Yes

Reviewer #2: No

Reviewer #3: Yes

5. Is the manuscript presented in an intelligible fashion and written in standard English?

Reviewer #1: Yes

Reviewer #2: Yes

Reviewer #3: Yes

Reviewer #1: The authors addressed all the comments very well and the manuscript can be accepted for publication.

Reviewer #2: Thank you for responding to my comments, however, in the stress-strain curve provided, authors claim, for 2% and 4% graphene-filled composites: a noticeable nonlinear region. I do not see any nonlinear behaviour before ultimate stress point. Please clarify this.

Reviewer #3: Dear Author,

The responses to all the reviewers’ queries have been addressed well by the authors. The quality of the paper is good. I agree to accept this article for publication

**Do you want your identity to be public for this peer review?** For information about this choice, including consent withdrawal, please see our Privacy Policy

Reviewer #1: No

Reviewer #2: No

Reviewer #3: **Yes:** Dr.R.Soundararajan, Professor - Mechanical Engineering, Sri Krishna College of Engineering and Technology, Coimbatore, Tamilnadu, India

---

## [Author Response · Author response to Decision Letter 2]

12 Dec 2025

Dear Editor and reviwer thank you for your time and kindly please expendit the peer review process of our manuscript

---

## [Decision Letter · Decision Letter 2]

29 Dec 2025

Effect of Graphene Addition on Tensile, Flexural, and Hardness Behavior of GFRP Composites

PONE-D-25-37781R2

Dear Dr. Syum,

We’re pleased to inform you that your manuscript has been judged scientifically suitable for publication and will be formally accepted for publication once it meets all outstanding technical requirements.

Kind regards,

Karthik Kannan, Ph. D.,

Academic Editor

PLOS One

Reviewers' comments:

Reviewer's Responses to Questions

**Comments to the Author**

Reviewer #2: All comments have been addressed

2. Is the manuscript technically sound, and do the data support the conclusions?

Reviewer #2: Partly

3. Has the statistical analysis been performed appropriately and rigorously?

Reviewer #2: Yes

4. Have the authors made all data underlying the findings in their manuscript fully available?

Reviewer #2: No

5. Is the manuscript presented in an intelligible fashion and written in standard English?

Reviewer #2: Yes

Reviewer #2: Although, I have accepted the manuscript, but, I am not convinced with the explanation that the behaviour for 2% and 4% is nonlinear in the stress-strain curve before UTS. I do not have access to the raw data.

**Do you want your identity to be public for this peer review?** For information about this choice, including consent withdrawal, please see our Privacy Policy

Reviewer #2: **Yes:** Manouchehr Salehi

---

## [Editor Report · Acceptance letter]

PONE-D-25-37781R2

PLOS One

Dear Dr. Syum,

I'm pleased to inform you that your manuscript has been deemed suitable for publication in PLOS One. Congratulations! Your manuscript is now being handed over to our production team.

Kind regards,

on behalf of

Prof. Karthik Kannan

Academic Editor

PLOS One